# Immune Checkpoint Inhibitors and RAS–ERK Pathway-Targeted Drugs as Combined Therapy for the Treatment of Melanoma

**DOI:** 10.3390/biom12111562

**Published:** 2022-10-26

**Authors:** Marta Morante, Atanasio Pandiella, Piero Crespo, Ana Herrero

**Affiliations:** 1Instituto de Biomedicina y Biotecnología de Cantabria (IBBTEC), Consejo Superior de Investigaciones Científicas (CSIC)—Universidad de Cantabria, 39011 Santander, Spain; 2Centro de Investigación Biomédica en Red de Cáncer (CIBERONC), Instituto de Salud Carlos III, 28009 Madrid, Spain; 3Centro de Investigación del Cáncer, Consejo Superior de Investigaciones Científicas (CSIC)—Universidad de Salamanca and IBSAL, 37007 Salamanca, Spain

**Keywords:** RAS–ERK, melanoma, inhibitors, immunotherapy

## Abstract

Metastatic melanoma is a highly immunogenic tumor with very poor survival rates due to immune system escape-mechanisms. Immune checkpoint inhibitors (ICIs) targeting the cytotoxic T-lymphocyte-associated protein 4 (CTLA4) and the programmed death-1 (PD1) receptors, are being used to impede immune evasion. This immunotherapy entails an increment in the overall survival rates. However, melanoma cells respond with evasive molecular mechanisms. ERK cascade inhibitors are also used in metastatic melanoma treatment, with the RAF activity blockade being the main therapeutic approach for such purpose, and in combination with MEK inhibitors improves many parameters of clinical efficacy. Despite their efficacy in inhibiting ERK signaling, the rewiring of the melanoma cell-signaling results in disease relapse, constituting the reinstatement of ERK activation, which is a common cause of some resistance mechanisms. Recent studies revealed that the combination of RAS–ERK pathway inhibitors and ICI therapy present promising advantages for metastatic melanoma treatment. Here, we present a recompilation of the combined therapies clinically evaluated in patients.

## 1. Introduction

Melanoma is unquestionably the most aggressive form of skin cancer. It generally arises due to the accumulation of genetic mutations in melanocytes, the pigment-producing cells in the skin, mostly as a consequence of overexposure to sunlight. Once the disease has extended from the initial lesion to become metastatic, the prognosis is very poor and the final outcome is fatal in the majority of the cases. The American Cancer Society estimates that there will be more than 106,000 cases and 7200 deaths in the United States caused by melanoma in 2021 [1]. With a historical perspective, annual melanoma incidence rates are escalating as rapidly as 4–6% [2].

Melanoma is characterized by being a highly immunogenic tumor [3]. Such a feature should induce adaptive immune responses in the organism, aimed at preventing tumor progression. For this, it is necessary that melanoma cells present adequate amounts of antigens, both qualitative and quantitatively, in order to trigger immune activation, instead of immune tolerance. This process depends on multiple factors pertaining both to the tumor cells themselves and the surrounding microenvironment [4]. In order to generate an effective anti-tumor immune response, seven steps, constituting what has been termed the Cancer–Immunity Cycle [5], must be enacted: (i) The release of cancer cell antigens; (ii) Cancer-specific antigen presentation by dendritic cells or antigen-presenting cells (APCs); (iii) Priming and activation of cytotoxic T-lymphocytes (CTLs) against the cancer-specific antigens that have been recognized as foreign; (iv) CTLs transportation to the tumor vicinity; (v) CTLs infiltration into the tumor; (vi) Recognition and binding to cancer cells by the CTLs; and (vii) The killing of the targeted cancer cells. 

In spite of their potential immunogenicity, melanoma cells have developed mechanisms of immune escape, based on attenuating the response of the tumor microenvironment [6]. This can be achieved by: (i) impeding an optimal activation of melanoma-infiltrating lymphocytes [7,8]; (ii) through the inhibition of CTLs’ function, either by the up-regulation of immune checkpoint ligands [9] or by stimulating the populations of immune suppressive cells such as myeloid-derived suppressor cells (MDSCs) [10] or regulatory T lymphocytes (Tregs) [11]; (iii) or by evoking CTLs’ death by apoptosis. In addition, other pro-tumorigenic effects, such as the stimulation of tumor angiogenesis and stroma remodeling [12], facilitate melanoma cells’ avoidance of the immune response.

## 2. Use of Immune Checkpoints Inhibitors in the Treatment of Melanoma

The interactions among tumor cells, APCs and T cells, as well as among T cells and the rest of the body’s cells, is orchestrated by a plethora of stimulatory and inhibitory molecules that regulate T cell activation [13]. Stimulatory molecules play a key role in activating the immune system. These molecules can regulate T cells’ responses by amplifying signals, carried out by co-stimulatory receptors, or counteracting signals, orchestrated by the co-inhibitory receptors [14]. So, in order to be active, the T cell has to be activated by two signals. First, the T-cell receptor (TCR) must recognize a part of a specific foreign epitope. The second signal is delivered by a co-stimulatory molecule. Without this last signal, the T cell is not fully active and it becomes anergic or dies. Once the T-cell activation is not needed, the co-inhibitor receptors come into play and curtail T-cell activation. Throughout all this process, the organism has immunological checkpoints, based on the action of inhibitory molecules, which prevent unwanted and harmful self-directed activities that lead to autoimmunity [15]. 

Immune checkpoints inhibitors (ICIs) are attracting enormous attention for the treatment of metastatic melanoma. The co-inhibitory receptors CTLA4 (cytotoxic T-lymphocyte-associated protein 4) and PD1 (programmed death-1) are two of the aforementioned inhibitory molecules, whose activation slows down the activity of the CTLs, resulting in an attenuation of the immune response against tumor cells [15]. However, melanoma cells have learned to utilize these inhibitory devices to their advantage, thereby evading immune destruction. For instance, the expression of CTLA4 in T cells is upregulated in melanoma cells, which provides that the T cells cannot be fully activated [16]. Moreover, these cells can also elude immune surveillance by expressing PDL1 (programmed death-ligand 1), the PD1 ligand, which ends up suppressing the T-cell function [17]. As such, during the last decade, the use of antibodies has been directed against the immune checkpoints such as CTLA4 and PD1, impeding the union between B7-CTLA4 and PD1-PDL1, respectively, (Figure 1). ICIs have yielded impressive clinical benefits for the treatment of metastatic melanoma, which has led to their approval both by the US and European drug agencies [18]. These ICIs have the goal of blocking specific immune checkpoint molecules in order to enhance intrinsic anticancer immunity [19].

CTLA4 is expressed on CD8+ and CD4+ T cells. Ipilimumab (anti-CTLA4 monocolonal antibody) was the first ICI to be approved by the FDA for the treatment of metastatic melanoma [20]. It is capable of preventing the CTLA4-induced CTL inhibition, which ends up with T-cell activation. Similarly, nivolumab and pembrolizumab target the interaction between PD1 with its ligands PDL1 (also known as B7-H1 or CD274) and PDL2 (also known as B7-DC or CD273) and have also obtained FDA approvals for the treatment of patients with unresectable or metastatic melanoma [21]. PD1 is found in the surface of T and B lymphocytes, natural killer cells and some myeloid populations. Upon its blockade by an antibody, its immunomodulatory function will be impaired, which will allow the T cells to continue and be active against the tumor [22]. Other antibodies against the ligand PDL1, expressed in the tumoral cells, are atezolizumab and durvalumab, which are also being tested in different clinical trials for the treatment of melanoma. Recently, a new clinical trial is incorporating a new anti-CTLA4 antibody named quavonlimab [23].

Before the appearance of antibodies against ICIs, patients with metastatic melanoma had a 5-year survival rate of 23% [24]. The therapeutic options consisted of antineoplastic chemotherapy drugs (such as dacarbazine) and high doses of interleukin-2 (IL-2), which resulted in severe adverse effects and low, overall survival (OS) benefit. With the implementation of ICIs, alone or in combination, the survival figures improved, with a rise in OS to 50% and response rates around 40% [25,26]. Immunotherapy provides long-lasting responses (more than 30 months) in almost one third of the patients. However, immune-related adverse events (irAEs) are often associated with such treatments. These include tissue-specific inflammatory responses, more often appearing associated with therapies against CTLA4 than to those targeting PD1 [27]. The most common ailments include pruritus, rash, nausea, diarrhea and thyroid disorders [28], but most of them are manageable. Contrarily, clinical trials have unveiled that the combination of anti-CTL4 and anti-PD1 therapeutics, despite exhibiting improved clinical benefits, displays a significant surge in irAEs grade 3 or 4 which, in some cases, calls for the discontinuation of the treatment [29,30].

Unfortunately, initial ICI-responder patients develop disease progression in a period of time. A clinical trial to test Ipilimumab efficacy in unresectable stage III or IV metastatic melanoma showed that 60% of patients maintained the response for at least 2 years [31]. However, the disease progressed in non-responder patients or patients that acquired resistance mechanisms. Similarly, a high percentage of patients treated with the PD1/PDL1 inhibitor, pembrolizumab, lose response over the time [32,33]. Immunotherapy-acquired resistance mechanisms are due to the evasion of immune recognition of tumoral cells. Among others, intratumor heterogeneity and low neo-antigens presentation in tumor cells [34,35], exclusion of T cells from the tumor microenvironment [36,37,38] and modulation of T-cell function in an immunosuppressive tumor microenvironment [39,40], such as presence of myeloid cells or low levels of oxygen, seem to be the cause of such evasion [41]. Presence of myeloid-derived suppressor cells contribute to CD8+ T-cell apoptosis [39], that can be avoid by blocking the Fas/Fas ligand pathway, therefore enhancing the immunotherapy efficiency [42]. Moreover, T-cell apoptosis can be triggered by the tumor antigen CD73, whose nucleotidase activity contributes to the tumoral immune evasion [40]. 

The molecular mechanism underlying resistance to ICIs remains elusive and limited, pointing to dysregulated cancer metabolism and epigenetic alterations as the drivers of immunotherapy escape. Recent study of the disease evolution over 9 years of a metastatic melanoma patient have shown some insights into the acquired resistance mechanisms [43,44]. Most of the tumor samples exhibited PTEN loss and genome duplication causing instability and aneuploidy. Moreover, few driver-alterations were identified, such as CDKN2A, epigenetic alterations and DNA-damage sensors’ dysregulation. However, cancer drivers such as BRAF or H/N/KRAS oncogenes were not found. The burden of immune cells in the tumor microenvironment was also diminished in resistant metastasis. Together with other studies, the common features during immunotherapy resistance are B-catenin activation [44,45,46], PTEN loss [37,47], lack of response to IFNg [48,49], depletion of tumor-specific neo-antigen presentation [49,50], genome instability [51,52], cell-cycle dysregulation [53] and epigenetic modulations [38,54].

## 3. The RAS–ERK Pathway

### 3.1. An Overview

The RAS–ERK pathway ranks amongst the most exhaustively studied signal transduction routes. RAS family GTPases operate as distributive hubs in the reception of incoming signals generated by extracellular stimuli and their subsequent allotment, through different downstream effector pathways, to the interior of the cell. Of these pathways, probably the best characterized is the one connecting RAS, specifically: HRAS, NRAS and KRAS, to the activation of ERK1/2 Mitogen-Activated Protein Kinases. This comprises a three-tiered signaling module sequentially linking MAPKKKs of the RAF family serine/threonine kinases (ARAF, BRAF and CRAF); MAPKKs dual-specificity kinases MEK 1 and 2; and MAPKs ERK 1 and 2, in a cascade of serial phosphorylation events, whereby the kinases at the different tiers are consecutively activated. Noticeably, data accumulated over two decades reveal that most of the constituents of the RAS–ERK pathway are capable of forming higher-order assemblies, particularly dimers [55]. As the last step in the cascade, once phosphorylated/activated, ERKs ultimately convey signals to an ample repertoire of substrates, localized at different subcellular localizations, which eventually translate the biochemical signal into the regulation of key cellular processes such as proliferation, differentiation, survival and motility, in addition to a wide variety of cell-specific activities [56,57].

Under resting conditions, unphosphorylated ERK1/2 are primarily cytoplasmic proteins. This is largely a consequence of their interaction with different types of proteins that serve as cytoplasmic anchors. These include: MEK1, certain protein phosphatases, and the cytoskeleton [58,59,60]. Upon phosphorylation, ERK1/2 lose affinity for their cytoplasmic partners and undergo a rapid redistribution throughout the cell, including a rapid translocation to the nucleus [61]. Many of the proteins identified as bona fide ERK1/2 substrates have turned out to be nuclear proteins. These include: nuclear structural components, transcription factors, nuclear receptors, other types of transcriptional regulators, chromatin constituents and proteins involved in cell-cycle regulation (for an extensive list of ERK1/2 substrates, see Yoon, 2006 [62,63]). In agreement, ERK1/2 tasks at the nucleus include the regulation of gene expression, growth factor-induced DNA replication [64], chromatin remodeling [65], and cell–cycle progression [66].

On the other hand, ERK1/2 functions outside of the nucleus have also been shown to be critical for essential biological processes. Among these are included: the control of cellular motility via the orchestration of cytoskeletal dynamics [67]; through the control of cell-to-matrix interactions by: integrin engagement [68]; integrin receptors’ affinity [69]; and M calpain-mediated adhesion [70]; the regulation of vesicular trafficking, such as endosomal traffic [71] and Golgi fragmentation [72]; and the control of protein synthesis, including ribosomal assembly [73] and protein elongation [74]. All this is achieved through the phosphorylation of nearly half of the 180 proteins identified thus far as ERK1/2 substrates, which happen to have an extranuclear localization. These include not only soluble cytoplasmic proteins but also constituents of the plasma-membrane, endomembranes, mitochondria and other organelles and the cytoskeleton [62]. Among these can be found proteins such as RSK1 [75], p70 S6 kinase [76], cPLA2 [77] and phosphodiesterase 4D (PDE4) [78]. Remarkably, whereas ERK1/2 exert their functions at the nucleus as monomers, ERK cytoplasmic activity is undertaken by ERK in dimeric form [56].

The dissection of the mechanistic details underlying the biochemical and biological functions of the RAS–ERK pathway, has evolved concomitantly with the gain of a great wealth of knowledge on the role that this route plays in cancer. Since the finding of HRAS as the first human oncogene [79,80], we have learned that the RAS gene family appears as the most frequently mutated in human cancer: 22%, 8.2% and 3.7% for KRAS, NRAS and HRAS, respectively [81]. Furthermore, to these figures must be added the cases in which oncogenic mutations are found in BRAF (22%) and those, far more infrequent, cases of mutational activation in MEK1/2 or ERK2 (<1%). Overall, over 30% of human neoplasias harbor oncogenic mutations in constituents of the RAS–ERK pathway [82]. However, significant as it is, this ratio fades when compared to the fractions that refer to specific tumors, where the proportion of cases harboring RAS–ERK pathway oncogenes is even greater. For instance, in pancreatic carcinoma KRAS appears mutated in 99% of the cases; meanwhile in thyroid cancer, oncogenic BRAF is detected in 40–60% of the cases [83]. In addition to this, the past three decades have witnessed the accumulation of a vast amount of data obtained by diverse gain- and loss-of-function genetic approaches, both in cellular and animal models, that unquestionably link the RAS–ERK pathway to tumor initiation, progression and dissemination [57]. As such, throughout this time colossal efforts have been devoted, both by academia and industry, to scrutinize this pathway in search of potential molecular targets for therapeutic intervention.

### 3.2. Immunological Impact of Targeting RAS–ERK Pathway

Cutaneous melanoma ranks amongst the tumor types with the highest rate of oncogenic mutations in members of the RAS–ERK pathway: 15–20% of melanomas harbor mutant NRAS, whereas oncogenic BRAF appears in 50–60% of the cases [84]. While, as mentioned, targeting the RAS–ERK pathway has proven an effective way for treating melanoma, such therapies have an important immunological impact on the tumor microenvironment (Figure 2) [85]. On the one hand, in vivo and clinical evidence support that targeting mutant BRAF increases antigen expression [86], MHCs class I and II [87] and PDL1 [86] expression in melanoma cells as well as an increase in CD8+ T-cell infiltrate in tumors of patients [88]. Moreover, the use of these inhibitors decreases the production of VEGF [89] and some immunosuppressive cytokines (IL-6, IL-8) [90] in tumor epithelial cells. In addition, inhibiting BRAF has also an impact on T cells such as an increase in T-cell exhaustion markers (such as TIM3 and PD1) [86], cytotoxicity [91], intratumoral Tregs [92], ratio CD8+/FoxP3/CD4+ [93] and clonality [94]. All of these impacts do not negatively affect the T-cell functions, since the drug targets a mutated variant of BRAF (V600E) that is not present in healthy cells. On the other hand, targeting MEK evokes a surge in antigen expression in melanoma cells [95] and MHC class I, II and PDL1 [96] expression in breast cancer cells. Moreover, by targeting melanoma cells with MEK inhibitors, a drop in the levels of VEGF and some immunosuppressive cytokines (IL-6, IL-10) is also observed [90]. However, when it comes to the effects on T cells, it has been observed that targeting MEK produces a decrease in T-cell proliferation [95,97] in normal human lymphocytes and in CD8/CD4/FOXP3 T-cell proliferation [98] in breast cancer. Finally, a decrease in the activation of antigen-specific CD8+ T cells has also been observed [97].

This difference in the behavior of the T cells with MEK inhibitors compared to BRAF inhibitors have led to different trials that combine both of the inhibitors with the potential to balance the overreacting effector cells to prevent exhaustion [99]. By combining both drugs, it is possible to restore antigen expression after progression on BRAFi alone, increase the infiltration of CD8+ T cell after progression on BRAFi alone [94] and raise CD4+ T-cell infiltration [100]. This rationale has been used in many clinical trials to assess the therapeutic potential of targeted therapies, as single agents or in combination, that will be described below.

#### 3.2.1. BRAF Inhibitors

BRAF inhibitors now constitute the main therapeutic approach for the treatment of metastatic melanoma. Mutations on BRAF, in particular V600 mutations, are associated with poor prognosis in some cancers such as metastatic melanoma [81]. According to clinical studies, the mutational status of BRAF determines the median survival, being 8.5 vs. 5.7 months for BRAF wild-type and mutant patients, respectively. The use BRAF inhibitors as a targeted therapy for BRAF-mutant cases has resulted in a significant improvement in the overall response (OR), progression-free survival (PFS) and overall survival (OS) rates [101] for the advanced disease. Depending on their mechanism of action, BRAF inhibitors can be classified as type-I and type-II inhibitors. Type-I BRAF inhibitors bind to the active conformation of BRAF, interacting with the ATP binding site, whereas type-II BRAF inhibitors interact with the inactive conformation of BRAF through an allosteric site [102].

The first FDA-approved RAF inhibitor, sorafenib, belongs to the type-II BRAF inhibitors and, in spite of a rather disappointing start, it showed little efficacy against BRAF-mutant melanoma—studies in other cancer types, such as renal cell carcinoma or hepatocellular carcinoma, have revealed a BRAF inhibition-independent antiproliferative and antitumoral effect, therefore the inhibition of tumor growth might be due to inhibition of other kinases [103,104]. Comparable to sorafenib, RAF265 is another type-II BRAF inhibitor shown to inhibit CRAF, BRAF wild-type, BRAF-V600E and to a lower extent other kinases [105]. It has been clinically evaluated for the treatment of melanoma in a Phase I trial which reported a partial response in 50% of the patients with high levels of hematologic toxicity [106].

As a consequence of their lack of selectivity for mutant BRAF, the use of type-II inhibitors resulted in intolerable toxicity. The advent of a second generation of agents, the ATP-competitive inhibitors selective for mutant BRAF [107], brought remarkable clinical benefits for patients, both in terms of disease-free evolution and overall survival [108,109]. Within the type-I inhibitors, the first small molecule with a high selectivity for mutant BRAF was vemurafenib (PLX4032) [110,111]. Since its identification from a Plexxikon structure-based drug discovery design [110], numerous studies have reported its potent activity as a selective inhibitor for BRAF V600 but not for wild-type BRAF cell lines [112], achieving cell-cycle arrest and apoptosis as well as tumor reduction in xenograft models of melanoma. In vitro studies have demonstrated that Vemurafenib exhibits an IC50 of 31 nM for mutant BRAF although it also exhibits inhibitory activity over CRAF with an IC50 of 48 nM, and other kinases at higher concentrations [107]. 

In clinical studies, the phase I trial BRAF in melanoma 1 (BRIM-1) showed a complete or partial tumor regression in 81% of patients with BRAF-V600E mutant metastatic melanoma [113]. This led to a phase II multicenter trial (BRIM-2) aimed at testing the efficacy and safety of vemurafenib in patients with advanced BRAF V600E-mutant melanoma who had received at least one prior therapy. The results indicated a reduction in overall response rate [114] (ORR 53%) that was confirmed in a larger phase III trial (BRIM-3), comparing vemurafenib (ORR 57%) vs. the control treatment dacarbazine [101,115]. Based on these results, the FDA [116] and the EMA [117] approved vemurafenib for the treatment of unresectable or metastatic melanoma. The toxicities found were consistent within the trials, with nausea, fatigue, skin-related toxicities and arthralgia being the most frequent adverse events [118].

Dabrafenib (GSK2118436) is a selective BRAFi that was approved in 2013 by the FDA as a single agent for the treatment of unresectable or metastatic melanoma in patients with the BRAF V600E mutation [119]. In preclinical studies [120,121], dabrafenib was shown to be specific for mutant BRAF, with an IC50 of 0.8 nM, although it can inhibit also wild-type BRAF (IC50 = 3.2 nM), CRAF (IC50 = 5 nM) and other kinases at higher concentrations. The inhibition of those kinases results in the blockade of cell proliferation in both melanoma cell lines and xenograft melanoma models [120]. In the first clinical trial (BREAK-1), dabrafenib showed a good overall response, in metastatic melanoma patients [122]. This was confirmed in a phase II trial (BREAK-2) [123], that also reported a better response in BRAF V600E tumors, as opposed to other BRAF V600 substitutions, such as V600K. Interestingly, another phase II trial focused on brain metastatic melanoma (BREAK-MB) [124] evaluating the effect of dabrafenib in previously treated and untreated BRAF V600E and V600K cases. It concluded that BRAF V600E patients responded better when they were not previously treated with other BRAFi. However, BRAF V600K patients showed a lower overall response, concomitantly with previous clinical trials, and the response was worse if those patients were not previously treated with other selective drugs. Finally, a phase III clinical trial (BREAK-3) without previous treatments, obtained an ORR of 50% and PFS of 5 months in patients treated with dabrefinib [125,126], demonstrating benefits against the positive control group treated with dacarbazine. The toxicities were manageable with some dose reductions being necessary and a few terminations due to adverse events [118].

The latest type-I inhibitor approved for the treatment of metastatic melanoma is encorafenib (LGX818), a selective and specific BRAF V600E inhibitor. To demonstrate BRAF V600E selectivity, a panel of mutant cell lines was examined for proliferation upon encorafenib treatment. BRAF V600E-mutant but not wild-type BRAF cell lines, exhibited a downregulation of cell proliferation (EC50 = 4 nM) and ERK phosphorylation (EC50 = 3 nM). This antiproliferative response correlated with the suppression of tumor growth in BRAF V600 melanoma-xenografted mice [124]. In a single agent phase I clinical study, encorafenib showed more potent and prolonged pharmacodynamic activity compared with vemurafenib and dabrafenib [127]. In the expansion phase, the overall response rate was higher for not previously treated patients than for those previously exposed to BRAF inhibitors. Further studies with encorafenib have shown that used in combination with the MEK inhibitor binimetinib [128] in metastatic BRAF-mutant melanoma cases, improves clinical response in terms of OS.

#### 3.2.2. MEK Inhibitors

The development of inhibitors targeting MEK1/2 kinases, has yielded the non-ATP-competitive allosteric inhibitors trametinib, cobimetinib [129] and binimetinib, which brought hope for non-BRAF-mutant melanoma cases. Remarkably, MEK inhibitors have also turned out to be more effective in BRAF- than in NRAS-driven tumors [130,131]. 

Thus, trametinib has been the only one to obtain FDA approval [132] as a single agent for the treatment of melanoma [133] (GSK1120212). In preclinical studies, trametinib was shown to reduce growth in both BRAF- and NRAS-mutant cell lines [134]. However, in xenograft melanoma models, trametinib was more effective in BRAF-mutant tumors [130,135]. In a phase I dose escalation trial for advanced solid tumors a dose of 2 mg was established. The phase II study in patients with metastatic BRAF-mutant cutaneous melanoma, previously treated with BRAF or other types of inhibitors, reported an ORR of 25% and median PFS of 4 months in the group without previous BRAFi therapy. Interestingly, patients previously treated with BRAF inhibitors had a worse response in terms of tumor size and progression-free survival, suggesting that BRAF inhibition-resistant mechanisms activate cancer-related reprogramming of the cell in such a way that MEK inhibitor sensitivity is lower. Later, in a phase III clinical study as a single agent for patients with BRAF V600E- or V600K-mutant melanoma, treatment with trametinib obtained an ORR of 22% and PFS of 4.8 months, compared to an ORR of 8% and a PFS of 1.5 months for the group treated with chemotherapy [136].

#### 3.2.3. Combination of BRAF and MEK Inhibitors

More recently, phase III trials have unveiled that the combined administration of MEK and RAF inhibitors significantly improves all of the parameters of clinical efficacy [137,138], which has led to FDA approving the use of trametinib/dabrafenib [136], cobimetinib/vemurafenib [139] and encorafenib/binimetinib [140] as the standard treatment for BRAF-mutant melanoma. It is worth mentioning that in a phase III clinical trial, the combination of encorafenib/binimetinib obtained an overall survival of more than 30 months for patients with the BRAFV600E mutation, a novelty for this kind of therapy. Furthermore, combined treatment with a new generation pan-RAF inhibitor and trametinib has also resulted in synergistic beneficial effects in NRAS-mutant melanoma cases [141], which has brought hope to those melanoma cases hitherto orphans of a treatment.

#### 3.2.4. Drawbacks in RAS–ERK Pathway-Targeted Therapy

Even though the RAS–ERK pathway small molecule inhibitors constitute a milestone in the treatment of cancer, all of them exhibit limitations that hamper their clinical performance. For example, BRAF inhibitors display an exquisite affinity towards mutant BRAF, thereby their highly specific, tumor-selective effects in cells harboring BRAF-V600E. On the contrary, MEK inhibitors cannot discriminate between the tumor and normal cells as they are not selective for MEK-mutant forms [142]. This results in toxic effects that limit their therapeutic use, both in dosage and treatment duration. 

A major problem for all types of treatments is the onset of resistance. With hardly any exceptions, the clinical response to RAF–ERK pathway inhibitors is short-lived, and relapses are the rule after about a year of treatment [143]. The mechanisms of resistance for BRAFi in patients with advanced-stage melanoma can be classified in three different ways. A primary or intrinsic resistance, that is characteristic of a non-response to treatment. An adaptive resistance, with an initial response or tolerance to non-mutant, early and reversible drugs. Finally, an acquired resistance with mutational drug tolerance, occurring late and irreversibly (nicely reviewed in Tian et al. [144]). The majority of the antitumoral agents trigger apoptosis-activating pathways, that is why the activation of anti-apoptotic molecules plays an important role during chemotherapy resistance. In this aspect, the key resistance targets are: Bcl-2 and associated antiapoptotic proteins, autophagy, necrosis, proteasome activation and epigenetic modulation among others [145]. The overexpression of Bcl-2 and related proteins that mediate intrinsic anti-apoptosis regulation, has caught attention from researchers in the hunt for specific small molecules to overcome the Bcl2-mediated resistance to chemotherapy. Moreover, some apoptosis-related proteins have been associated with melanoma sensitivity to BRAF and MEK inhibitors, such as the Bcl2-interacting killer (BIK) protein that mediates on BRAF inhibitor effectiveness [146] and Survivin that contributes to melanoma cells survival by inhibiting apoptosis [147]. 

Interestingly, histone deacetylase (HDAC) proteins function as an apoptosis suppressor in melanoma cells, being part of the BRAF-inhibitor resistance mechanisms. By a poorly understood event, the combination of BRAF inhibitors and pan-HDAC inhibitors provides clinical benefits to metastatic melanoma patients [148]. However, not only apoptosis-related proteins are involved in anti-apoptosis-mediated melanoma resistance; long non-coding RNA (lncRNA) also participated in such regulation. A recent study has revealed that the lncRNA TSLNC8 expression is downregulated in BRAFi resistant melanoma cells by inhibiting apoptosis. Importantly, the overexpression of TSLNC8 re-sensitizes the BRAFi resistant melanoma cells [149]. 

From a molecular point of view, the reinstatement of ERK activation is the hallmark of many BRAF/MEK inhibitor resistance mechanisms. Furthermore, the clinical efficacy of a treatment correlates with the capacity of the therapeutic agent to prevent ERK activation, and a sustained and potent ERK inhibition is a requisite for attaining a durable response [150]. Unfortunately, natural selection results in tumor adaptation when ERK activity is reinstated through distinct molecular mechanisms. These include: the amplification of genes encoding for upstream receptors or RAS–ERK pathway constituents; down-regulation of RAS negative regulators; the appearance of BRAF splice isoforms; overexpression of CRAF; RAF isoform switch; activation of alternative MAPKKKs; and the appearance of MEK or ERK mutants [151,152,153].

## 4. Rationale for Combining RAS–ERK Pathway-targeted Therapy and Immunotherapy

In the absence of a clear biomarker that can determine which patients will benefit the most from each type of treatment [154], emerging evidence supports the idea of combining RAS–ERK pathway-targeted therapy and immune checkpoint inhibitors may be promising for the treatment of metastatic melanoma (Figure 3). The rationale for this combination relies on the fact that treatment with BRAF and MEK inhibitors results in a more immuno-responsive tumor microenvironment [155]. In general, patients treated with RAS–ERK signal inhibitors show short-term benefits and high ORR, while those treated with immunotherapy achieve longer term benefits, though with a lower ORR [156]. 

Studies in mouse models have provided useful information about the effects of the combination of both approaches. For example, tumor-infiltrating T cells surge when BRAF inhibitors and anti-PD1 antibodies are combined, prolonging survival and slowing tumor growth [157], as revealed by a BRAFV600E/PTEN-/- syngeneic tumor graft immunocompetent mouse model. With a similar methodology, Deken and colleagues also tested the combination of BRAFi or MEKi therapy with anti-PD-1 immunotherapy and observed increased tumor reduction in response to the combinatorial therapy, in comparison to RAS–ERK signal-targeted therapy alone [158]. 

Others used a mouse model of syngeneic BRAF(V600E)-driven melanoma. With this model, they tested the additional effect of trametinib (MEKi) on the combination of immunotherapy with dabrafenib (BRAFi), showing an increase in the antitumor effect in comparison to the targeted therapy or the immunotherapy alone [92]. Using the same model, Moreno and colleagues added to the triple therapy (dabrafenib, trametinib and immunotherapy based on PD1 blockade) the antibodies targeting CD137 (4-1BB) and/or CD134 (OX40) [100], observing an improvement in the antitumor activity with the addition of these new antibodies to the triple therapy. These observations were detected when the antibodies were added separately to the triple therapy. 

Taken together, preclinical studies demonstrated a beneficial effect of the combination of RAS–ERK pathway-targeted therapy with immunotherapy, providing the rationale for clinical trials in patients.

### Clinical Trials Combining MEK/BRAF Inhibitors with Immunotherapy

Based on the aforementioned preclinical data, there are several ongoing clinical trials combining MEK and BRAF inhibitors with ICIs for melanoma treatment (Table 1) [159], but to date, only a few results have been published. Moreover, not every trial ends up being completed. For example, a phase I clinical trial (NCT01400451) aimed to test the combination of vemurafenib with ipilimumab in patients with metastatic BRAF-mutant melanoma, had to be terminated due to the high incidence of hepatotoxicity [160]. Conversely, another phase II clinical trial (NCT01673854) tested vemurafenib followed by ipilimumab, with no hepatotoxicity reported [161]. 

Due to the overall high toxicity evoked by therapies using antibodies against CTLA4, these have been substituted by those directed against PD1 and PDL1, to be tested in combination with BRAF-MEK inhibitors, based on the underlying principle that, when used as a monotherapy, PD1 and PDL1 antibodies display lower toxicities than those against CTLA4 [162]. A first example is the phase Ib study (NCT01656642) using atezolizumab (an anti-PDL1) in combination with vemurafenib or vemurafenib plus cobimetinib in patients with BRAF-V600 metastatic melanoma [163]. The study reported an unconfirmed RECIST V1.1 response in 85.3% of the cases.

Another phase I trial (NCT02130466 or KEYNOTE 022) with an anti-PD1 (pembrolizumab) plus dabrafenib and trametinib for BRAF-mutant advanced melanoma, reported an unconfirmed overall response rate of 67% with a high frequency of grade 3–4 treatment-related adverse events [164]. Based on those results, a phase II clinical trial (part 3) was performed where patients were randomized to receive concurrent dabrafenib and trametinib with pembrolizumab or placebo [165]. Surprisingly, results based on the overall response rate were better for the placebo group, while the progression-free survival improved for those patients treated with the immunotherapy. Immune-mediated adverse events occurred in a higher percentage in the cohort treated with pembrolizumab (43% vs. 13%). Finally, there are two new parts of this trial (parts 4 and 5), that aimed to use pembrolizumab plus trametinib for patients with solid tumors or BRAF wild-type melanoma, with results still pending publication. Other studies (NCT03149029, NCT02858921) are also exploring the combination of pembrolizumab with dabrafenib and trametinib.

Durvalumab, an anti-PDL1, has also been used in combination with dabrafenib and trametinib in patients with both BRAF-mutated and BRAF-wild-type advanced melanoma [166] in a phase I clinical trial (NCT02027961). The patients were divided in three different cohorts: (A) BRAF-mutated patients treated with dabrafenib, trametinib, and durvalumab; (B) BRAF-wild-type patients treated either with trametinib and durvalumab; (C) sequential trametinib followed by durvalumab. The early results of 50 patients showed a better response rate for the group treated with the combination of both BRAF and MEK inhibitors plus the immunotherapy rather than just one of the inhibitors or the sequential treatment. Moreover, the treatment-related toxicities for this first cohort were observed in 39% of the patients [166]. 

Promising results from the phase II clinical trial of triplet combination of nivolumab with dabrafenib and trametinib (TRIDeNT) in patients with PD1 naïve or refractory BRAF-mutated metastatic melanoma have also been published [167]. After flowing a six patients’ safety run-in with no observed dose-limiting toxicities, 27 patients were treated with the combination, and more than 90% of the patients reached overall response rate with a median progression-free survival of 8.5 months [167].

There are also some promising results from phase III clinical trials. Atezolizumab, vemurafenib and cobimetinib were used in a phase III clinical trial (NCT02908672 or IMspire150) [168] as the first-line treatment for unresectable advanced BRAFV600 positive mutation. This is the first phase III clinical trial aimed at evaluating an immune checkpoint inhibitor combined with BRAF plus MEK inhibitors in patients with advanced BRAFV600 mutation-positive melanoma. A total of 514 patients were enrolled and randomly assigned to atezolizumab plus vemurafenib and cobimetinib (atezolizumab group, *n* = 256) or placebo plus vemurafenib and cobimetinib (control group, *n* = 258). After a median follow up of 18.9 months, the progression-free survival was higher for the atezolizumab group in comparison to the control group. Moreover, both groups obtained a similar safety profile. Based on these results, atezolizumab has been granted approval for use in combination with cobimetinib and vemurafenib for patients with BRAF V600 mutation-positive unresectable or metastatic melanoma [169].

Another phase III clinical trial (NCT02967692 or COMBI-I) compared an anti-PD1 (spartalizumab or PDR001) in combination with dabrafenib and trametinib with the targeted therapies alone. Recently, it has been reported that the trial did not meet its primary endpoint of investigator-assessed PFS for patients treated with the investigational triplet therapy, although an important trend was seen in favor of the triple therapy arm. Serious treatment-related adverse events occurred in 23.2% vs. 11% (triplet therapy vs. dabrafenib and trametinib alone correspondingly) [170].

The IMspire170 (NCT03273153) was another phase III study where patients were randomized to receive cobimetinib plus atezolizumab or pembrolizumab alone. However, the primary outcome (progression-free survival) was not reached [171].

## 5. Conclusions

Despite all efforts, metastatic melanoma remains an incurable disease for many patients. There is strong evidence for the beneficial clinical effects that BRAF and MEK inhibitors have in metastatic melanoma, increasing tumor immunogenicity, but the onset of resistance mechanisms makes them a temporary solution in many cases. Clinical data show that the combination of targeted therapy and immunotherapy can be beneficial, especially triple combinations of BRAFi+MEKi+ICIs, but there are still many open questions that must be resolved. The election of the antibody to act on the immune checkpoint is one of them. Despite its benefits in monotherapy, it has been demonstrated that antibodies anti-CTLA4 display more toxicity than those targeting PD1/PDL1 association. Nevertheless, the optimal timing regarding when to initiate the treatment or what kind of agent is the best option (targeted therapy vs. immunotherapy) are also open enquiries. Retrospective data show that the results are similar regardless of the sequentiality [172,173], but further studies need to be completed. In this regard, the study SECOMBIT (Sequential Combo Immuno and Target Therapy, NCT02631447) [174], a phase II study aimed at evaluating the sequentiality of two different combinations and to evaluate which of both approaches is better, is still too preliminary to be applied in the clinic. 

Another pressing issue is the need for a strategy to stratify which patients will benefit the most from each treatment. An urgent need in this matter is the biomarking. Some of the best-studied biomarkers are: PD1/PDL1 expression, tumor mutational burden, gene expression harboring significant mutations (for example BRAF) or tumor infiltrating lymphocytes (TILs). However, the clinical data available indicate that these biomarkers are still insufficient to fairly predict an optimal response to ICIs.

## Figures and Tables

**Figure 1 biomolecules-12-01562-f001:**
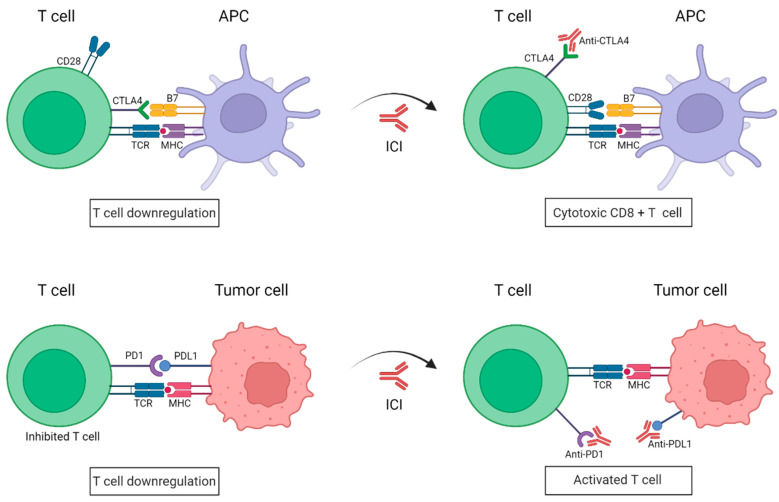
**Schematic representation of immune checkpoint blockade by antibodies anti-PD1 and anti-CTLA4.** Under normal conditions, the binding of PD1 expressed on activated T cells to its ligand PDL1 on tumor cells causes T-cell exhaustion. CTLA4 and CD28 are found in T cells and they compete with each other to bind their ligand B7 present on APCs. CTLA4 has a higher affinity to B7 than CD28 and transmits an inhibitory signal to T cells, whereas CD28 transmits a T-cell stimulatory signal. In the presence of ICIs, anti-PD1/PDL1 antibodies impede the union of PD1 to its ligand PDL1, resulting in a killed tumor cell; whereas anti-CTL4 antibodies work by blocking the inhibitory B7-CTLA-4 signaling, allowing T cells to be active. APC: Antigen-presenting cell; TCR: T-cell receptor; MHC: major histocompatibility complex; PD1: programmed death-1; PDL1: programmed death-ligand 1; CTLA4: cytotoxic T lymphocyte antigen. (Created with BioRender; www.biorender.com, access date: 20 October 2022).

**Figure 2 biomolecules-12-01562-f002:**
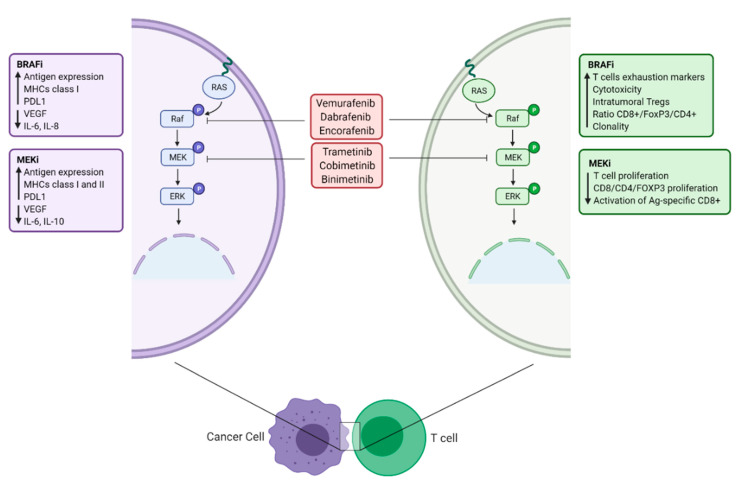
**Immunological effects on tumor microenvironment of BRAF and MEK inhibitors.** Representation of the immunological consequences of targeted therapy with BRAFi (vemurafenib, dabrafenib and encorafenib) and MEKi (trametinib, cobimetinib and binimetinib) in a cancer cell (purple) and a T cell (green). Adapted from Bedognetti et al [85]. (Created with BioRender; www.biorender.com, access date: 20 October 2022).

**Figure 3 biomolecules-12-01562-f003:**
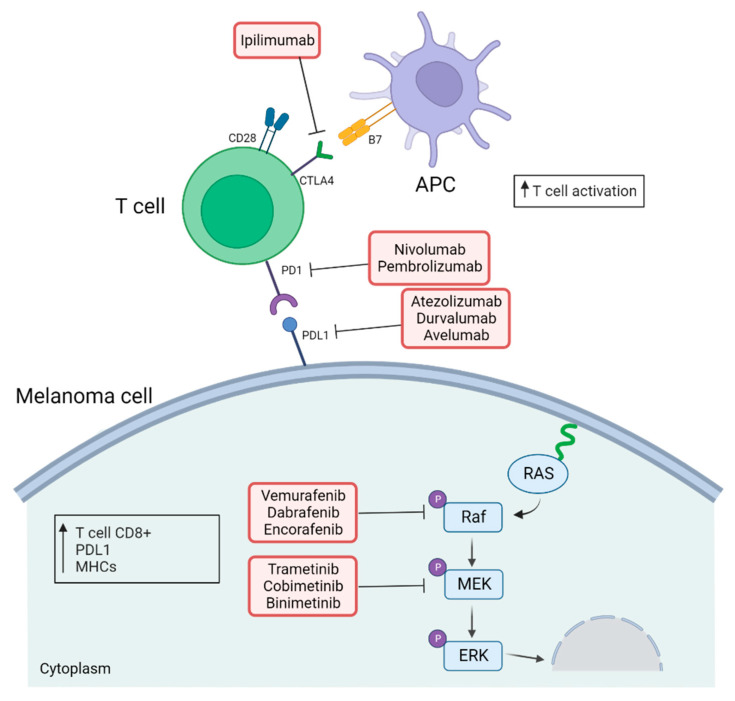
**Effects of combination of immune checkpoint inhibitors and RAS–ERK pathway-targeted drugs.** Representation of the effect of anti-CTLA4 and anti-PD1/PDL1 antibodies along with inhibitors of the RAS–ERK pathway to treat patients with metastatic melanoma (created with BioRender; www.biorender.com, access date: 20 October 2022).

**Table 1 biomolecules-12-01562-t001:** Clinical trials combining BRAF and MEK inhibitors with ICIs in melanoma.

NTC Number	Phase	Targeted Therapy	Immunotherapy	Conditions	Status
NCT04835805	I	Belvarafenib, Cobimetinib	Atezolizumab	Advanced Melanoma	Recruiting
NCT04722575	II	Cobimetinib, Vemurafenib	Atezolizumab	Resectable Melanoma	Recruiting
NCT01400451	I	Vemurafenib	Ipilimumab	Advanced melanoma	Terminated
NCT01673854	II	Vemurafenib	Ipilimumab	Advanced melanoma	Completed
NCT02200562	I	Dabrafenib	Ipilimumab	Stage III/IV Melanoma	Terminated
NCT02130466	I/II	Dabrafenib, Trametinib	Pembrolizumab	Advanced Melanoma	Completed
NCT03149029	II	Dabrafenib, Trametinib	Pembrolizumab	Metastatic Melanoma	Active, not recruiting
NCT02858921	II	Dabrafenib, Trametinib	Pembrolizumab	Stage III Melanoma	Active, not recruiting
NCT02625337	II	Dabrafenib, Trametinib	Pembrolizumab	Metastatic Melanoma	Unknown status
NCT02027961	I	Dabrafenib, Trametinib	Durvalumab	Advanced melanoma	Completed
NCT02910700	II	Dabrafenib, Trametinib	Nivolumab	Stage III/IV Metastatic Melanoma	Recruiting
NCT02908672	III	Cobimetinib, Vemurafenib	Atezolizumab	Metastatic or Unresectable Melanoma	Active, not recruiting
NCT03273153	III	Cobimetinib	Atezolizumab, Pembrolizumab	Advanced Melanoma	Completed
NCT02967692	III	Dabrafenib, Trametinib	Spartalizumab	Metastatic or Unresectable Melanoma	Active, not recruiting
NCT04657991	III	Encorafenib, Binimetinib	Pembrolizumab	Melanoma	Recruiting
NCT04655157	I/II	Encorafenib, Binimetinib	Nivolumab, Ipilimumab	Metastatic Melanoma	Recruiting
NCT04511013	II	Encorafenib, Binimetinib	Nivolumab, Ipilimumab	Melanoma	Recruiting
NCT03554083	II	Cobimetinib, Vemurafenib	Atezolizumab	Stage III Melanoma	Recruiting
NCT03235245	II	Encorafenib, Binimetinib	Nivolumab, Ipilimumab	Unresectable Stage III/IV Melanoma	Recruiting
NCT03175432	II	Cobimetinib	Atezolizumab, Bevacizumab	Untreated Melanoma Brain Metastases	Recruiting
NCT02902029	II	Cobimetinib, Vemurafenib	Atezolizumab	Metastatic or Unresectable Melanoma	Active, not recruiting
NCT02818023	I	Cobimetinib, Vemurafenib	Pembrolizumab	Advanced Melanoma	Terminated
NCT02303951	II	Cobimetinib, Vemurafenib	Atezolizumab	Malignant Melanoma	Terminated
NCT01940809	I	Dabrafenib, Trametinib	Nivolumab, Ipilimumab	Stage III/IV Metastatic Melanoma	Terminated
NCT01767454	I	Dabrafenib, Trametinib	Ipilimumab	Metastatic or Unresectable Melanoma	Completed
NCT01656642	I	Cobimetinib, Vemurafenib	Atezolizumab	Metastatic Melanoma	Completed
NCT03178851	I	Cobimetinib	Atezolizumab	Malignant Melanoma	Completed
NCT02902042	I/II	Encorafenib, Binimetinib	Pembrolizumab	Malignant Melanoma	Completed
NCT02224781	III	Dabrafenib, Trametinib	Ipilimumab, Nivolumab	Metastatic Melanoma	Active, not recruiting

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
