# Peer review of "Immune Checkpoint Inhibitors and RAS–ERK Pathway-Targeted Drugs as Combined Therapy for the Treatment of Melanoma"

_biomolecules, 2022, doi:10.3390/biom12111562_

Round 1

Reviewer 1 Report

The review article titled "Immune checkpoint inhibitors and RAS-ERK pathway-targeted drugs as combined therapy for the treatment of melanoma" by Morante et al., is an interesting article describing the use of inhibitors of the Ras and MEK opathways as well as immune checkpoint inhibitors in overcoming resistance of melanoma cells. 

The authors have addressed various aspects of immune evasion, and have used adequate number of related references (which can be improved). A major shortcoming of the manuscript is that there no section on apoptosis resistance and apoptosis-associated molecules involved in resistance and how to manipulate their expression levels to bypass (overcome) resistance. 

All of these inhibitors affect the apoptotic machinery of the melanoma cells (intrinsic and extrinsic), therefore a major section should be dedicated to this issue

Author Response

We appreciate Reviewer´s comments and suggestion. We agree, apoptosis is one of the key events during chemoresistance and indeed more information was needed in the manuscript. Now, we have extended the sections for resistance and include some data about apoptosis. However, we believe there are very complete and detailed revision of the matter in the literature, and the focus of our manuscript is not on the therapy evasion mechanisms. That is why we have not include a whole section about apoptosis related chemoresistance, and we have edited and ameliorated the content of the corresponding sections

Reviewer 2 Report

1. The authors are better to emphasis the aspect of Anti-PD1 and Anti-CTLA4 as a medication. In other words, please show us how Anti-PD1 and anti-CTLA4 effect tumor cells as treatments as schema.

2. Can PDL-1 and PD1 bind even in the presence of Anti-PD1? Judging from Figure 1, it looks that way.

3. From Figure 1, it is difficult to understand the function of Anti-CTLA4.

4. The authors are needed to show the combined therapy (immune checkpoint inhibitors and RAS-ERK pathway-targeted 2 drugs) as a schema. 

Author Response

1. The authors are better to emphasis the aspect of Anti-PD1 and Anti-CTLA4 as a medication. In other words, please show us how Anti-PD1 and anti-CTLA4 effect tumor cells as treatments as schema.

We thank the reviewer for the comment. Now, we have edited the figure caption to emphasis anti-PD1/CTLA4 function over tumor cells.

2. Can PDL-1 and PD1 bind even in the presence of Anti-PD1? Judging from Figure 1, it looks that way.

From the representation in figure 1, we wanted to indicate the binding site of Anti-PD1 antibody. Now, we have corrected the caption in order to clearly show its mode of action.

3. From Figure 1, it is difficult to understand the function of Anti-CTLA4.

We thank the reviewer´s comment. Anti-PD1 and anti-CTLA4 functions were not clear enough in the figure, probably due to a consequence of making schema simple for better visualization. We hope that the edited indications in the caption help in the understanding of their functions.

4. The authors are needed to show the combined therapy (immune checkpoint inhibitors and RAS-ERK pathway-targeted 2 drugs) as a schema. 

We appreciate this comment and agree on the need of a combined therapy containing schema. We have included a Figure 3 with the representation of both ICIs and BRAF/MEK inhibitors as combined therapy.

Round 2

Reviewer 1 Report

Although the authors have briefly discussed Bcl-2 member BIK and its relevance to apoptosis and the signaling pathways, however, this section is inadequate for such a comprehensive review. A separate and detailed section on apoptosis-associate gene products and their possible regulation by inhibitors need to be included. 

Author Response

We strongly disagree. The main purpose of this review is to focus on the advantages of combined therapy for melanoma treatment, not to provide a detailed compilation of the mechanisms underlying chemoresistance. In our context, devoting a major section to apoptosis resistance would completely distort the purpose of our review and, furthermore, would completely unbalance it. We should then review all the events occurring during chemoresistance and not only apoptosis.

Reviewer 2 Report

We would like the schema itself to be changed to fit the authors statement.

Author Response

We have now modified Figure 1 to make it more clear.